# Microplastics, Additives, and Plasticizers in Freshwater Bivalves: Preliminary Research of Biomonitoring

Giulia Cesarini [1,2,*] , Fabiana Corami [3,4] , Beatrice Rosso [4] and Massimiliano Scalici [1]

1. Department of Sciences, University of Roma Tre, Viale G. Marconi 446, 00146 Rome, Italy; massimiliano.scalici@uniroma3.it
2. Water Research Institute, CNR-IRSA, L.go Tonolli 50, 28922 Verbania, Italy
3. Institute of Polar Sciences, CNR-ISP, Campus Scientifico, Ca' Foscari University of Venice, Via Torino 155, 30172 Venezia, Italy; fabiana.corami@cnr.it
4. Department of Environmental Sciences, Informatics, and Statistics, Ca' Foscari University of Venice, Via Torino 155, 30172 Venezia, Italy; beatrice.rosso@unive.it
* Correspondence: giulia.cesarini@uniroma3.it; Tel.: +39-0657336355; Fax: +39-065733632

**Abstract:** Microplastics are widespread in freshwater environments and could impact these ecosystems. Bivalves are freshwater organisms that are particularly exposed to microplastic contamination. Therefore, in this preliminary study, the accumulation of microplastics, plasticizers, and additives in the freshwater bivalves *Anodonta cygnea* was investigated through active biomonitoring. Specimens bought commercially were exposed in three rivers in Central Italy for different exposure times: short (1 month) and long (3 months). The gills and the gastrointestinal tract (GIT) were analyzed separately to evaluate the possible uptake and ingestion of particles via Micro-FTIR. For the first time, small microplastics (SMPs, 5–100 μm), plasticizers, additives, and other micro-litter components, e.g., natural and non-plastic synthetic fibers (APFs), were identified in the bivalve *A. cygnea*. The most abundant polymer in the gills (94.4%) and in the GITs (66.1%) was polyamide, which had the highest concentration in each river. A decrease in SMPs' abundance was observed over time in the gills in each river, while the abundance in the GIT increased. Compared to polymers, a greater variety of APFs was observed in rivers. The APFs changed during the time of exposure and between different rivers more evidently than polymers, allowing for a clearer identification of the possible sources. These results highlighted the plastic pollution caused by SMPs using freshwater bivalves as sentinel organisms and the need to further investigate the additives that can be proxies of the presence of microplastics in the environment and biota.

**Keywords:** freshwaters; microplastics; bivalve organisms; environmental exposure; gills; gastrointestinal tracts; biological uptake; nylon; rayon





## 1. Introduction

The issue of microplastic pollution, regarding occurrence, source and possible impacts, has raised the interest of many scientists and its presence in the environment is well-documented [1–6]. These ubiquitous pollutants are widespread in freshwater, which can be affected by anthropogenic factors (e.g., proximity of urban centers, low efficiency of wastewater treatments plants (WWTPs), use of sewage debris for fields), while natural factors (e.g., wind, storms, floods) contribute to their dispersion in the aquatic environment [7–9]. Aquatic organisms can actively ingest microplastics (MPs < 5 mm) [10–13] and these particles may exert adverse effects on individual, cellular or molecular levels [14–16]. The most common effect of MPs is the reduction in food uptake, which is probably due to false food satiation and particularly observed in combination with other contaminants [17–19]. Indeed, MPs can also represent vectors of environmental pollutants and pathogen microorganisms, increasing the ecological risk due to their adsorptive capac-

ity [20,21]. In addition, the presence of additives in plastic materials to enhance polymer properties poses several chemical risks to biota [22].

The techniques used to investigate MP concentration in environmental matrices may provide responses limited in time as water and sediment are affected by several environmental perturbations that can modify the level of plastic contamination very quickly [23]. Moreover, in the case of sediment, the processing analyses are more complicated than those used for most biota due to the complexity of the soil matrix [23]. In this sense, the use of bioindicators can provides an integrated assessment of plastic pollution [24–26]. Among the proposed bioindicators, bivalves can be used as valuable sentinel organisms indicating the plastic level in the aquatic environment [23,27,28]. The large sizes of several species belonging to the Unionidae family, such as *Anodonta cynea* (Linnaeus, 1758), provide sufficient material for chemical analysis even when their population density is low [29].

*Anodonta cygnea* is a freshwater bivalve indigenous in several countries of Europe and Asia [30]. This freshwater mussel is one of the largest bivalves occurring in permanent rivers with slow currents, lakes, and pools; it has also been observed in canals, drainage, and dam reservoirs [31]. Bivalves *A. cygnea* inhabit water bodies characterized by fertile bottom sediments and by high concentrations of dissolved oxygen [32,33]. This species can filter several liters of water (2.6–2.9 L/h), and, for this reason, it is selected as a natural filter in aquaculture [34,35]. Given the feeding strategy and their large dimensions, mussels could uptake MPs and in particular small microplastics (SMPs, <100 μm) similar to the size of seston, as well as additives, plasticizers, and other micro-litter components, e.g., natural and non-plastic synthetic fibers (APFs) [12]. Despite the multiple advantages of using bivalves as bioindicators of MP pollution, freshwater bivalves are poorly investigated, especially in this field [36].

To fill this gap, the aim of this study was to assess the uptake and ingestion of SMPs and APFs in *A. cygnea* by analyzing the gills and the gastrointestinal tract (GIT) separately. Bivalves in three rivers of central Italy were exposed for one and three months, investigating the quantity, size, and shape of SMPs and APFs. In this research, *A. cygnea* was evaluated as a sentinel organism that can be employed for the biomonitoring of MPs pollution in freshwater environments. Specifically, our study employs a novel method for investigating all present polymers without denaturation and analyzes the gills and gastrointestinal tract separately, allowing for a more thorough evaluation of uptake and ingestion. These aspects represent important advancements in the analysis of plastic pollution and contribute to a deeper understanding of the impacts of MPs on freshwater bivalves.

## 2. Material and Methods

### 2.1. Experimental Design and Environmental Exposure of Bivalves

Eighteen specimens of *A. cygnea* of similar size (Table S1) were purchased from commercial breeding in Italy, and the species has been confirmed via the morphology of the shell (Aldridge, 1999) [37]. Three specimens were analyzed immediately after being bought in order to collect SMPs data regarding environmental pre-exposure (T0). The other specimens were exposed to environmental conditions in three different rivers (Marta, Aniene, and Sacco) for different exposure times. The rivers are located in the Lazio Region, and the investigated sites are located in their potamal tracts (Figure 1). The physicochemical parameters of rivers are shown in Table S2.

The sites are surrounded by different land uses; the Aniene River is predominantly characterized by urban use, while the Marta River and the Sacco River are characterized by agricultural use. The Aniene River flows entirely within the Lazio Region (99 km) and is the second largest tributary of the Tiber River, crossing a large part of the city of Rome, which is characterized by a high degree of anthropization [38]. The investigated site is located in the urban park of Aniene Valley in the east part of Rome city, where the Aniene receives many WWTPs discharges. The Marta River flows into the Tyrrhenian Sea after a course of 54 km, and the site of investigation is located in the cultivated countryside near Tarquinia, a small city in the north of the Lazio Region. Finally, the Sacco River, a tributary

of the Liri River, flows for 87 km in the territory of Frosinone, in the south of Lazio, and is surrounded by an agricultural and industrial context. The investigated site is located near Colleferro city, where the river becomes polluted due to the discharges of many industries in the area [39].

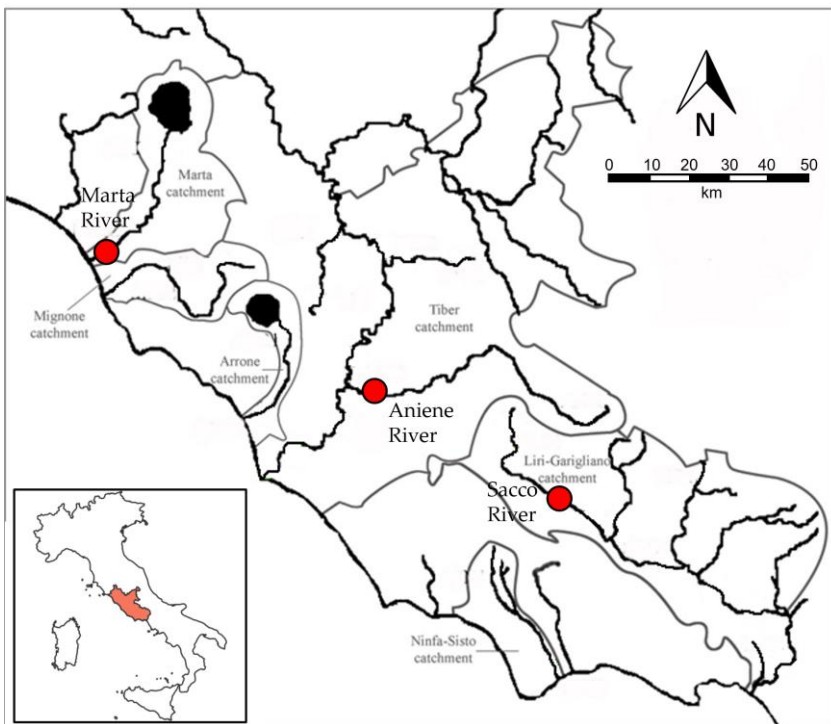

**Figure 1.** Location of the investigated sites (red circles) in three different rivers of the Lazio Region (Central Italy) where specimens of *Anodonta cygnea* were exposed for short and long time periods. For all figures, colors are only availabled in the online version.

Five specimens of *A. cygnea* were placed in a homemade iron cage that allowed flow-through conditions in each river. A stone that allowed the trap to sink and settle on the superficial bottom (5 cm) was inserted in each cage. The cages were attached to a riparian tree using a rope of about 30 m. In this way, the traps can reach the middle part of the river to avoid the riverbanks and any lowering of the water level that would cause the death of specimens. Specimens were collected from the cages to evaluate the results obtained of environmental exposure in the short time (ST) period after one month (August), and long time (LT) period after three months (August–October). To evaluate plastic accumulations during ST and LT, two and three individuals, respectively, were analyzed. Despite the small number of samples investigated, *A. cygnea* provides sufficient material for analysis given its large size.

### 2.2. Quality Assurance/Quality Control (QA/QC)

To avoid airborne plastic contamination, the shells were opened, and the gills and the GIT were removed using dissecting scissors and steel tweezers under a laminar flow hood wearing cotton lab coats and nitrile gloves. The stored samples were sent to the Institute of Polar Sciences (CNR-ISP, Venezia-Mestre, Italy) for MP analysis. The specimens of the same sampling date and river were pooled in one sample. The pooled samples were analyzed in replicates (n = 3) for gills and GITs. All pre-analytical operations were conducted in a plastic-free clean room ISO7, where the atmospheric pressure, humidity, temperature, and particle pollution are controlled. Only glass and steel objects were employed. All glassware (including filtration apparatus) was washed with a 2% solution of Contrad 70 (Decon Laboratories Limited, Hove, UK) and rinsed several times with ultrapure water. Then, all glassware and inox steel tools were decontaminated with a 50% ($v/v$) solution of

methanol (for HPLC ≥ 99.9% Sigma Aldrich, Merck, Darmstadt, Germany) and ethanol (absolute, for HPLC ≥ 99.8% Sigma Aldrich, Merck, Darmstadt, Germany) and allowed to dry under a fume hood in the cleanroom. Reagent blanks (i.e., ultrapure water, filters and reagents) and procedural blanks were performed. The aluminum oxide filters (ANODISC filters, Supported Anopore Inorganic Membrane, 0.2 µm, 47 mm, Whatman™, Merck, Darmstadt, Germany) were placed in decontaminated Petri glass dishes after filtration using decontaminated steel tweezers and covered with aluminum foil. Contamination was also avoided when transferring filters from the cleanroom to the micro-FTIR laboratory, stored in glass Petri dishes and covered with clean aluminum foil. During the micro-FTIR analyses, each filter was quickly put on a stage and then covered with the protection of the instrument. During all operations, cotton lab coats and nitrile gloves were worn.

### 2.3. Dissection of Gills and Gastrointestinal Tracts

After the sample collection, bivalves exposed to environmental conditions were transported to the laboratory and stored frozen (−20 °C) in aluminum foil divided according to the sampling location until dissection. The wet gills and the wet GITs were weighed (0.1 g, Kern 440-47, Germany) and stored separately in a sterile glass container with 80% ethanol pooled per different samplings and control. For each specimen, the maximum shell length, width, and height were measured (Table S1).

### 2.4. SMPs and APFs Extraction Procedures and Analysis via Micro-FTIR

The extraction and purification procedure were employed according to the method developed by [12] which resulted in the minimization of any possible polymer degradation, and the method's yield was >90%. The particles were not further denatured, even polyamide, which can be denatured with temperatures ≥ 55 °C, as is often employed in extraction procedures [12,13,40].

Briefly, after the $H_2O_2$ (30%-RPE for analysis-ACS-Reag.Ph.Eur.-Reag.USP, Carlo Erba) digestion of gills and GITs, the digested samples were filtered whit a vacuum pump Laboport® (VWR International, Milan, Italy) and the quantification and simultaneous polymer identification of the filters were conducted using a micro-FTIR Nicolet iN10 infrared microscope (Thermo Fisher Scientific, Madison, WI, USA). Each filter was analyzed in transmittance mode with the Particles Wizard section of Omnic™ Picta™ software (https://knowledge1.thermofisher.com/Molecular_Spectroscopy/Molecular_Spectroscopy_Software/OMNIC_Family/OMNIC_Picta_Software, accessed on 10 July 2023), which also enables the collection of each particle's length and width through its imaging [12,41]. The analysis parameters are reported in the Supplementary Materials (Figure S1). The quantification was performed via microscopic counting: at least 14 count fields (1.8 mm² each) were randomly chosen with no overlapping on the surface of the filter (area 1734.07 mm²). The spectral background was acquired on a clean point of each count field, and each spectrum was identified via comparison with a suite of reference libraries (see Table S3), and spectra with a match percentage (match%) ≥ 65% were accepted. Only the SMPs and AFPs characterized by the optimal match of identification were quantified. The total number of SMPs and AFPs per gills or GITs and the weight of SMPs per specimen were then calculated according to the equations reported in [12] (see also formulae in Supplementary Materials).

SMPs were related to geometric solids as a result of their aspect ratio. The aspect ratio (AR) is the ratio between the maximum length (L) and the maximum width (W) of the smallest rectangle (bounding box) enclosing the particle chosen. When the AR ≤ 1, the particle is considered spherical; when the AR ≤ 2, it is considered elliptical/elongated; when the AR ≥ 3, it is considered cylindrical; and in case of AR ≥ 9, it is considered a fiber [12].

*2.5. Statistical Analysis*

The abundance and distribution of SMPs and APFs and their weights are expressed as the number of particles per gram wet weight (g ww). The normality distribution of the dataset was evaluated using the Shapiro–Wilk test. In the case where the normality hypothesis was rejected, a non-parametric test was performed. The chi-squared test ($\chi^2$) was used to compare the concentrations of SMPs and APFs in the same river at different times of exposure (ST vs. LT) in both the gills and GITs. The degree of freedom (df) for this test was also determined. To analyze the differences in SMPs and APFs concentrations among different sites, the non-parametric Kruskal–Wallis test (H) was used. This test is used to compare datasets that do not follow a normal distribution. If this test showed significant differences among the sites, then Dunn's post hoc test was applied for multiple comparisons. The statistical significance level was set at *p*-value (*p*) < 0.05; non-significance *p*-value are reported as ns. Statistical analyses were performed using GraphPad Prism software (version 8.0.1).

We hypothesized that the concentration of SMPs and APFs would increase over time in the GIT and decrease in the gills. Additionally, we expected that in urban rivers, such as the Aniene, the concentration of SMPs and APFs would be higher compared to that of rivers with lower levels of urbanization.

## 3. Results and Discussion

*3.1. SMPs Uptake and Ingestion by Anodonta cygnea*

This study represents the first record of SMPs and APFs in the freshwater bivalve *A. cygnea*, highlighting that this species can ingest these particles in the environment. This is a preliminary study that, in the future, would need to use more individuals to confirm the results found. During the environmental exposure, SMPs were found in all the samples. Overall, 18 plastic polymers, and their acronyms, were identified (Table S4). These analyses confirm the presence of SMPs in bivalves, both in the gills and GITs of the organisms exposed in the three rivers. Figure 2a shows the abundance of the SMPs accumulated in the gills and GITs (n SMPs/g ww), while Figure 2b shows the relative weight of SMPs (μg SMPs/g ww). The weight shows that although the particles are abundant, they are contaminants at the level of μg/g. There are polymers that are more abundant in terms of particle number but not as regards weight because they can be small particles or low-density polymers.

The average value of SMPs found in the T0 sample was the highest (1953 SMPs/g ww). Other studies confirmed that the MP concentrations were higher in the aquaculture environment compared to the natural environment [42,43]. Concerning the environmental samples, the highest average value of SMPs, considering gills and GITS together, was found in the Aniene River (1615 SMPs/g ww), followed by the Marta River (1445 SMPs/g ww), and, finally, the Sacco River (980 SMPs/g ww). In the literature, a correlation was found between the degree of urbanization and MP concentration [44–46]. Indeed, the Aniene is a river flowing within the city of Rome and the one surrounded by the highest rate of urbanization and population density, both of which are considered predictors of plastic pollution [47,48]. In addition, research conducted in the potamal tracts of rivers located in the Lazio Region, including some of the same sites investigated in this study, found the highest concentration of plastic along the riverbank of Aniene River [49].

The differences in SMPs concentration in gills between different times of exposure (ST vs. LT) were found to be significant in each river: Aniene ($\chi^2$ = 752.2; df = 5; *p* < 0.0001), Marta ($\chi^2$ = 422.2; df = 8; *p* < 0.0001), and Sacco ($\chi^2$ = 599.1; df = 5; *p* < 0.0001). Significant differences were also found in polymer concentrations in GITs: Aniene ($\chi^2$ = 4831; df = 6; *p* < 0.0001), Marta ($\chi^2$ = 2300; df = 5; *p* < 0.0001), and Sacco ($\chi^2$ = 526.3; df = 5; *p* < 0.0001). In gills, the abundance of SMPs significantly decreased over time in each river, suggesting that gills act as a zone of interchange between the medium and organism, while in the GIT, the abundance of SMPs significantly accumulated over time (Figure 2). There is another study that investigated the accumulation in *Unio pictorum* bivalves exposed to sewage

treatment plant effluents over time, finding an increase in MPs concentration after 28 days of exposure; however, the study did not analyze the organs separately [50].

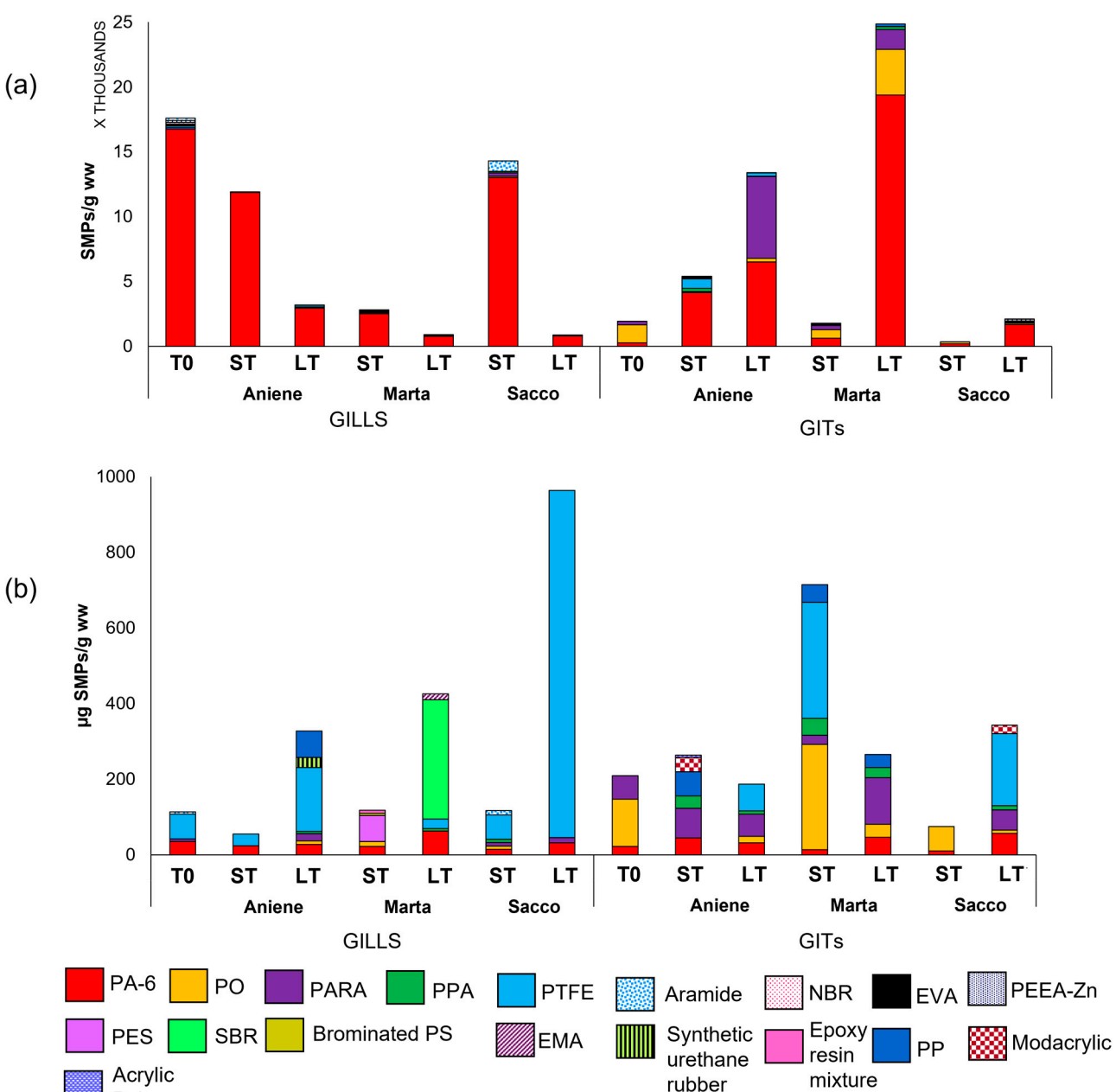

**Figure 2.** Concentration (**a**) and weight (**b**) of different small microplastic polymers (SMPs) in the gills and GITs of bivalves *A. cygnea* exposed in three rivers (Aniene, Marta, Sacco) for a short time (ST) and long time (LT). T0 refers to the individuals of pre-exposure. For each polymer, the abbreviation is reported: acrylic polymer = ACRYLIC POLYMER; aramide = ARAMIDE; brominated polystyrene = BROMINATED PS; epoxy resin mixture = EPOXY RESIN MIXTURE; ethylene methyl acrylate copolymer = EMA; ethylene vinyl acetate = EVA; modacrylic = MODACRYLIC; nitrile-butadiene rubber = NBR; polyamide = PA-6; polyarylamide = PARA; polyester = PES; polyethylene acrylic acid copolymer-zinc salt = PEEA-Zn; polyolefin = PO; polyphthalamide = PPA; polypropylene = PP; polytetrafluoroethylene = PTFE; styrene–butadiene Rubber = SBR; synthetic urethane rubber = SYNTHETIC URETHANE RUBBER.

Gills are used to filter nutrients and eliminate debris, by exhaling siphon, including SMPs, producing feces and pseudofeces. A study highlighted that in *Corbicula fluminea*

exposed to nanoplastics, the production of feces and pseudofeces increased, suggesting the enhancement of mechanisms for the release of non-edible particulate matter [51]. Another explanation for the reduction in SMPs in gills over time may be the decreasing filtration rate caused by physiological or internal restraints due to MPs [19]. The filtration of bivalves is mainly influenced by environmental conditions, which determine the filtration rate [34]. In our case study, the investigated sites were characterized by similar chemical–physical conditions and were in optimal oxygen and pH conditions; therefore, the filtration efficiency remains unchanged between different rivers. Through filtration and excretion, bivalves impact the cycle of nutrients in freshwater ecosystems [52], including the environmental concentrations of MPs. It is important to consider that all the substances and particles, including MPs, expelled in the water column by bivalves become part of the trophic chain [53].

In addition to the MPs expelled, those accumulated in the bivalve can also pose a threat to the organism itself and its predators. Indeed, several laboratory studies highlighted the toxic effects of MP exposure in different species of freshwater bivalves, such as tissue damage, inflammatory response, intestinal damage, protein modulation and neurotoxicity [17,54–56]. In particular, the MP uptake may affect the reproduction process of Unionid individuals, as these bivalves incubate their larvae in the brood sacs composed of gill filaments and septum [53,57]. Furthermore, the consumption by predators of bivalves that have accumulated MPs can lead to biomagnification phenomena along the trophic chain [58,59].

Once the MP uptake by gills has occurred, the SMPs can be transferred and accumulated in different organs, as has been verified by laboratory studies. In our case study, we found an increasing accumulation of SMPs in GITs compared to gills over time. Indeed, [60] found that on mantles and gills there was a small number of SMPs (~100 μm), while in digestive glands and gonads, there was a high concentration. Similarly, other studies identified the highest concentrations of SMPs (32–250 μm) in the intestinal parts [16,61]. Since laboratory studies have shown a constant depuration of MPs by bivalves, this means that the investigated rivers are in a state of continuous plastic pollution and the SMPs remain bioavailable [62]. This hypothesis should be confirmed by carrying out studies on the surrounding water and sediment matrices as well, especially by investigating the surficial sediment in which MPs were found to be more similar to those found in bivalves [27,63].

Given the diversity of polymers found in each river, the bivalve *A. cygnea* represents a suitable sentinel organism to highlight plastic pollution in freshwater systems. The uptake in the field reflects the bioavailability and environmental concentration of SMPs in freshwater systems [27,60]. Polymers with a diverse range of densities were found, from PP (density = 0.905 g cm$^{-3}$) to PTFE (density = 2.2 g cm$^{-3}$) (Table S4). In Figure S1, some spectra with the highest match of identification are reported.

The differences observed in the distribution of polymers in the gills between different rivers were found to be non-significant (H = 5.132; *p* = ns). The same result was found for the polymer distributions in GITs, as no significant difference between samples was found (H = 7.237; *p* = ns). Indeed, in all sites, the most abundant polymer both in gills (94.4%) and GITs (66.1%) was polyamide (PA-6), known commonly as nylon. Therefore, the distribution of polymers in different rivers was similar as PA-6 is predominant in the SMPs composition. The extraction procedure of SMPs used in this research allowed the researchers to identify polymers with a high accuracy, enabling the detection of PA-6, which can be easily lost due to high temperatures or aggressive treatments [12,13,40,64,65]. PA-6 is very common in fishing nets and fish tackles employed in bivalve and fish farms [66]; a very high concentration was found in T0 (16,742 SMPs/g ww) from the specimens taken directly from the commercial breeding. Moreover, PA-6 is widely employed in fabrics for clothes and carpets and can be released from washing machines or originate from agricultural employment and transported via leaching or by the wind [40,67–70]. PA-6 was therefore ubiquitous and widely spread both in sites with more urban contexts, such as the Aniene River, and agricultural contexts, such as the Marta and Sacco rivers. Moreover, in other

studies investigating aquatic biota, both marine and freshwater, nylon was the polymer most abundantly found [13,71–73].

All average size values of plastic particles found were <100 μm. Indeed, it has been highlighted from laboratory studies that bivalves mainly incorporate MPs of a smaller size (17–88 μm) as these are easier to digest compared to larger ones, which are often rejected by the organisms [53]. The average length and width of SMPs found in gills were $52.68 \pm 7.38$ μm and $26.55 \pm 3.90$ μm, respectively, while in GITs, these were $58.38 \pm 7.02$ μm and $30.52 \pm 3.47$ μm, respectively. In Table S5, the values of SMPs' sizes for each site are shown. Despite the size of the different rivers being very variable, thus reflecting the specific conditions of pollution, overall, SMPs are larger in the GITs than the gills, probably because the larger particles are retained and accumulated, while the smaller ones are easier to expel. The SMPs' sizes were found to be very similar to the sizes of nutrients and microalgae typically ingested by these freshwater bivalves. Therefore, it is probable the SMPs are mistaken for micronutrients and uptaken by the gills are then accumulated in the GITs [12,17,71]. Since the average size of SMPs found in the gills was very diverse, especially compared to T0, this means that there was a complete replacement. Therefore, bivalves are good representative organisms of the conditions of environmental disturbance of a specific site filtering plastics of different sizes.

Microplastic uptake has been highlighted in other species of bivalves [27,62,74,75], proving that these filter feeders are excellent bioindicators of MPs, reflecting the variability of plastic pollution in freshwaters. Comparing the accumulation of SMPs in *A. cygnea* in literature is impossible since this study represents the first record. Moreover, the comparison between this result and the results obtained from other research in bivalves is hindered by the scarcity of studies that investigate the concentration of SMPs by analyzing the organs separately and using proper techniques for polymer identification. However, considering other bivalve species and different methodologies employed, lower concentrations were found. In other species belonging to the Unionidae family, *Anodonta natine* and *Unio pictorum*, the concentrations were found to be 20.6–37.7 mps/individual and 0–9 mps/individual, respectively [50,76]. In contrast, considering species of smaller sizes, such as *Corbicula fluminea* (0.3–4.9 MPs/g ww; [27]) or *Dreissena polymorpha* (0.03–0.23 items/individual; [77]), the concentration of MPs decreased. Indeed, body size was found to be a relevant parameter for the possible MP quantities ingested, increasing with larger sizes [11,62]. Moreover, the number of particles ingested is also affected by their size (Sendra et al., 2021). The average size of SMPs found in gills and GITs of this study was smaller than the sizes found in other species, such as *Corbicula fluminea* and *Limnoperna fortune*, in which the dominant sizes were 500–1000 μm and 250–1000 μm, respectively, and lower concentrations of MPs ingested were found [27,78]. However, the comparison of concentrations between different bivalve species is speculative as all variables affecting the filtration rate should be taken into account.

Overall, the SMPs' most common shape was the ellipse (70%), followed by the sphere (15%) and cylinder (15%; Figure 3a). A fiber shape was occasionally found. Specifically, the ellipse was the most abundant shape found in each river (Figure 3b). Other studies have also found the ellipse to be the most abundant shape [12,13], while many others report fiber as the most common shape [27,63,76,77]. This may be due to the different methods of considering the shapes of MPs, but it is fundamental to understand that, in nature, there are more elongated/irregular shapes than perfect spheres. These types of shapes can be accumulated by bivalves in different organs and are more difficult to expel [78]. In laboratory studies, MP spheres are often used as they are easier to obtain, but it is necessary to further investigate the shapes that mainly interact with biota in the environment [79,80].

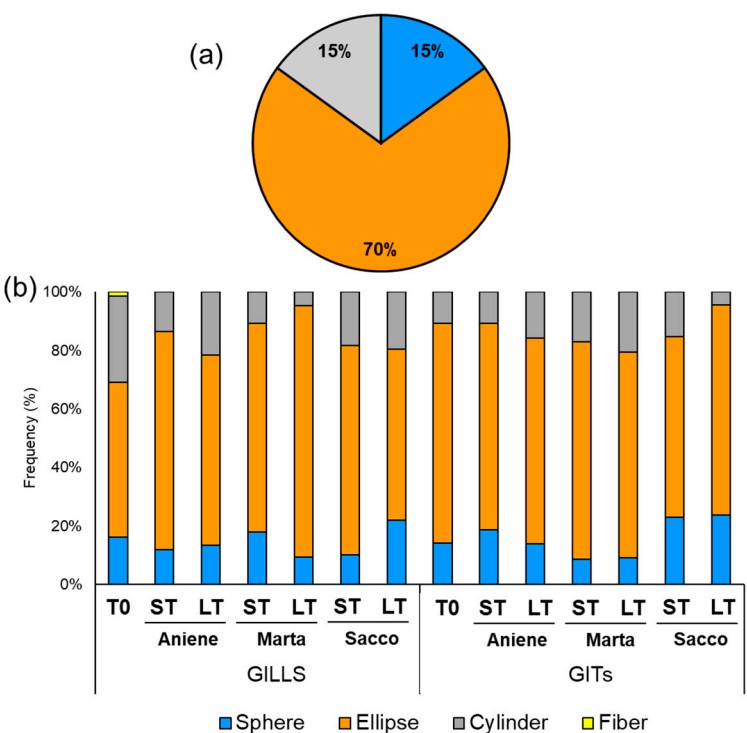

**Figure 3.** Frequency (%) of SMPs shapes total (**a**) and found in gills and GITs from different rivers (**b**). Sphere (AR $\leq$ 1), ellipse (AR $\leq$ 2), cylinder (AR $\geq$ 3) or fiber (AR $\geq$ 9).

### 3.2. AFPs Uptake and Ingestion by Anodonta cygnea

During the environmental exposure, 34 APFs and their acronyms were identified (Table S6). The abundance (n APFs/g ww) of the APFs accumulated by *A. cygnea* in gills and in GITs is shown in Figure 4.

The differences in APFs' concentration between different times of exposure (ST vs. LT) in each river were found to be significant: Aniene ($\chi^2$ = 819.3; df = 15; $p$ < 0.0001), Marta ($\chi^2$ = 2624; df = 7; $p$ < 0.0001), and Sacco ($\chi^2$ = 16,040; df = 7; $p$ < 0.0001). Significant differences between times of exposure were also found in APFs' concentrations in GITs: Aniene ($\chi^2$ = 19,807; df = 10; $p$ < 0.0001), Marta ($\chi^2$ = 21,706; df = 4; $p$ < 0.0001), and Sacco ($\chi^2$ = 9585; df = 2; $p$ < 0.0001). Although there is a significant difference in APFs' concentration between the ST and LT analyses of each river, there is no clear trend relating to the decrease or increase over time in gills and GITs as observed for polymers.

Among the APFs, silk (47.57%) was the most abundant observed in the gills overall. Silk is a natural component, and its presence can be related to one produced by bivalves [81] and other aquatic invertebrates, such as caddisflies and dipterans [13]. Following this, in gills, N-(2-ethoxyphenyl)-n-(2-ethylphenyl)-ethanediamide (NNE) (41.60%) and varox (5.35%) were very abundant overall. NNE is an additive functioning as a light stabilizer employed in linear low-density polyethene polymers intended for repeat food contact use [82]. Varox is a vulcanizing additive to polymerize resins and obtain tyres, tapes and flexible tubes in rubber and plastic (Yue et al., 2006). In GITs, the most abundant APFs was rayon (53.15%) overall. Following this, NNE (22.44%) and hombitan $TiO_2$ (13.63%) were very common. Rayon is a non-plastic synthetic fiber composed of cellulose regenerated with caustic soda to obtain viscose or cellophane and has been found in different habitats [83]. Commercial $TiO_2$ is a color additive used as a pigment or filler in plastics, paints, paper, foods, ceramics, and pharmaceuticals, and it is also a sunscreen [84]. Nanoparticles of $TiO_2$ were found to inhibit growth and cause direct physical effects on algae *Phaeodactylum tricornutum* [85].

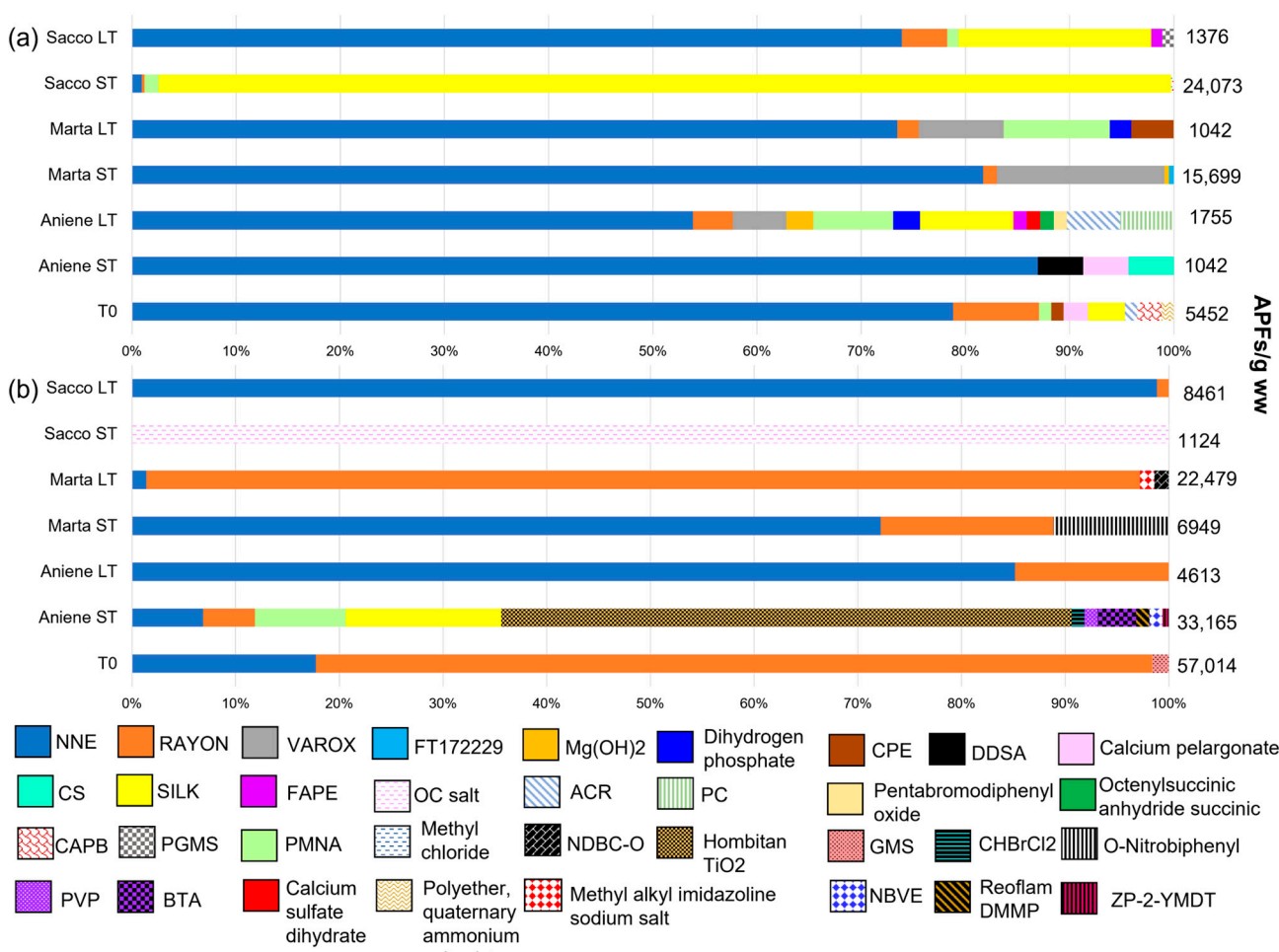

**Figure 4.** Concentrations in percentage (%) and total of different additives, plasticizers, and other micro-litter components (APFs) in *A. cygnea* (**a**) gills and (**b**) GITs exposed in three rivers (Aniene, Marta, Sacco) for short time (ST) and long time (LT). T0 is referred to the individuals of pre-exposure. For each additive, the abbreviation is reported: 50% active glycerol monostearate in polyethylene carrier = GMS; benzotriazole = BTA; Bromodiclorometahne = CHBrCl₂; butyl vinyl ether = NBVE; calcium pelargonate = CALCIUM PELARGONATE; calcium stearate = CS; calcium sulfate dihydrate = CALCIUM SULFATE DIHYDRATE; chloroalkyl phosphate ester = CPE; co-coamidoproyl betaine = CAPB; dodecenylsuccinic anhydride = DDSA; hombitan TiO₂ = HOMBITAN TIO₂; Magnesium hydroxide = Mg(OH)₂; methyl alkyl imidazoline sodium salt = METHYL ALKYL IMIDAZOLINE SODIUM SALT; methyl chloride = Methyl chloride; N-(2-ethoxyphenyl)-n-(2-ethylphenyl)-ethanediamide = NNE; nichel-dibuthyldithiocarbarmate = NDBC-O; Octadecanoic acid, calcium salt = OC SALT; octenylsuccinic anhydride succinic = OCTENYLSUCCINIC ANHYDRIDE SUCCINIC; O-nitrobiphenyl = O-NITROBIPHENYL; pentabromodiphenyl oxide = PENTABROMODIPHENYL OXIDE; phosphate ester fatty acids blend = FAPE; Poly(N-methyl acrylamide) = PNMA; polyether, quaternary ammonium salt mixture = POLYETHER, QUATERNARY AMMONIUM SALT MIXTURE; polyvynilpirollidone = PVP; propyl zithate = ZP-2-YMDT; Propylene carbonate = PC; propylene glycol monostearate = PGMS; rayon = RAYON; reoflam dmmp = REOFLAM DMMP; Silk = SILK; sodium polyacrylate, polyacrylic water = ACR; stearamidopropyldimethyl-hydroxyethylammonium-dihydrogen = DIHYDROGEN PHOSPHATE; triphenyl methane 4,4′,4″-triisocyanate in ethyl acetate = FT172229; varox 231 xl = VAROX.

Overall, the differences observed in the APFs' distribution in the gills resulted to be non-significant (H = 10.13; *p* = ns). Regarding the APFs' distributions in GITs, there was a significant difference between the samples (H = 22.19; *p* = 0.0011). The multiple comparison

using Dunn post hoc analyses revealed significant differences between T0 and Aniene ST ($p = 0.0409$), Aniene ST and Aniene LT ($p = 0.0056$), Aniene ST and Marta ST ($p = 0.0280$), Aniene ST and Sacco ST ($p = 0.0010$), and Aniene ST and Sacco LT ($p = 0.0047$). The APFs changed during the time of exposure and between different rivers more evidently than polymers, enabling a clear diversification of the possible sources. In particular, the Aniene River was found to be the most diverse among the rivers studied regarding the types of APFs found, as it is mainly characterized by additives related to urban WWTPs rather than an agricultural discharge, as in the case for the Marta and Sacco rivers.

Specifically, in the Aniene River, the most common additives were $TiO_2$ and NNE; additionally, only in this site, calcium pelargonate, an anionic surfactant used in lacquers, pharmaceuticals, plastic, and DDSA, another surfactant, were found in gills after a ST of exposure, probably due to the presence of several ditches, discharge and WWTPs [38]. In October, after a LT of exposure, there was an increase in APFs in gills related to wastewaters, including calcium sulfate dihydrate, used for water treatment, pharmaceuticals, insecticides, and plaster, and propylene carbonate, used in cosmetics, personal care products, detergents, and degreasers [82]. Moreover, in the GITs, several APFs only found in the Aniene River and related to WWTP were found, such as $TiO_2$ and polyvynilpirollidone (PVP).

In the Marta River, styrene butadiene rubber (SRB), a polymer mainly used to produce tyres, was found in August (ST) in the gills and varox was found both in August and October, highlighting a relationship between exposure times. The APFs in the Sacco River changed widely over time in the GITs, suggesting that there was an exchange between the bivalves and the surrounding medium. In this river, rayon and silk were very abundant, while cocamide, an additive used in cosmetics, was present in smaller quantities. However, although the Sacco River is mainly characterized by SMPs and AFPs, which suggests agricultural sources, there are also contributions from industrial activities, such as methyl chloride, used in the production of methylcellulose, butyl rubber, octadecanoic acid, and calcium salt (OC salt), used as an ingredient for paper collation and metal stearates. In fact, in this area, there are paper factories, many chemical industries and landfills, in addition to the agricultural use of the territory [86].

APFs can be toxic to biota as polymers, and since they have been less studied, their effects can be underestimated or completely unknown [82]. Additionally, the toxicological effects of a polymer depend on the chemical additives employed and the toxicity of a polymer without the additives is much lower [87]. Therefore, it is important that future studies analyze their bioavailability, accumulation, and toxicity. Studies on additives and plasticizers conducted to date have shown their toxicity on different aquatic organisms [22,88]. Moreover, impacts on human health have been identified, such as breast cancer, apoptosis, and genotoxicity [89].

## 4. Conclusions

For the first time, native bivalves *A. cygnea* were used as suitable model organisms for investigating the freshwater pollution of SMPs and APFs. It is worth mentioning that, in this study, the separation of the organs, gills and GITs was carried out, which allowed the researchers to investigate the uptake and ingestion of SMPs and APFs. This approach must be followed both in field and in laboratory studies to achieve a thorough understanding of the uptake, ingestion, and accumulation phenomena of plastics. A highlight of this research is the fact that we were able to determine the polymers and additives that are accumulated in the bivalves and, consequently, their bioavailability; thus, we suggest these be investigated in laboratory studies to analyze the toxicological effects. The analytic method used permitted the simultaneous determination of SMPs and APFs present in bivalves with high efficiency, such as polyamide, a polymer that is easily denatured using high temperatures and aggressive treatments, which was found in high quantities. The analysis of APFs enables a higher diversification of the possible sources than that of SMPS because these are more related to a specific usage. The high number of particles found may be due to the synergy of the method used and the size of the bivalves.

The gills seem to act as a zone of interchange with the medium and the number of particles decreased over time, while the GITs accumulate particles, increasing the concentration over time. Studies of active biomonitoring, as in this case, are very useful because these allow researchers to obtain comparable data due to the availability of the same quantity of organisms collected from different sites. Investigating the shape of MPs in freshwater environments is important to understand which shape should be focused on during laboratory experiments to reproduce as closely as possible the environmental conditions and analyze the possible effects on the organism. To confirm the observed trend of increased SMPs in GIT and decreased SMPs in gills, we recommend conducting further research with a larger number of organisms and longer exposure times. Additionally, analyzing SMPs and APFs in water and sediment matrices can provide a better understanding of the issues of bioaccumulation and bioavailability. Further investigation of APFs is needed as they can be considered tracers of the presence of microplastics in the environment and biota.

**Supplementary Materials:** The following supporting information can be downloaded at: https://www.mdpi.com/article/10.3390/w15142647/s1. Table S1. Biometric features (length, width, height) and weight of *Anodonta cygnea* collected from aquaculture (T0) and exposed in three different rivers for short term and long term. Table S2. Physicochemical parameters of river water at the three sampling sites. Table S3. List of reference libraries employed for the analysis via MicroFTIR, software PARTICLE WIZARDS, Omnic™ Picta™. Equations S1. Equations employed to obtain the total number of SMPs per gills or GITs and the weight of particles per specimen. Table S4. List of the polymers identified and quantified in the specimens of *A. cygnea* exposed in 3 different rivers for short term and long term, including T0. For each polymer the abbreviation and average density (g cm$^{-3}$) are reported. Figure S1. Some examples of the best FTIR spectra of polymers collected on the SMPs observed in gills and GITs of *Anodonta cygnea*. (A) Nylon or polyamide (PA), (B) polyolefin (PO) and (C) polyaryamide polypropilene (PP). The FTIR spectrum of the polymer collected is in red, the spectrum of the polymer present in the suite of reference libraries employed is in black. Spectral range of 4000–1200 cm$^{-1}$, 100-mm step size scanning (spatial resolution) at 100–100 mm aperture, and 32 co-added scans at the spectral resolution of 4 cm$^{-1}$. Optimal Match % of polymers was ≥65%. Table S5. Microplastic particles size measured in μm as length (L, the longest dimension) and width (W, perpendicular to the length) found in gills and GITs. Table S6. List of the additives, plasticizers, and other micro-litter components identified and quantified in the specimens of A. cygnea exposed in 3 different rivers for short term and long term, including T0. For each additive, the abbreviation and average density (g cm$^{-3}$) are reported.

**Author Contributions:** G.C.: Conceptualization, data curation, formal analysis, investigation, writing—original draft; writing—review and editing. F.C.: data curation, formal analysis, investigation, methodology, supervision, writing—review and editing. B.R.: data curation, formal analysis, investigation, methodology, writing—review and editing. M.S.: funding acquisition, conceptualization, resources, supervision, visualization, writing—review and editing. All authors have read and agreed to the published version of the manuscript.

**Funding:** This investigation was supported by funds of the Ministry of Education, University and Research for the base research individual activities, and by the Grant of Excellence Departments, MIUR-Italy (Article 1, Paragraphs 314–337, Law 232/2016).

**Data Availability Statement:** The datasets used and/or analyzed during the current study are available from the corresponding author upon reasonable request.

**Acknowledgments:** Thanks to the regional authority Roma Natura for authorizing this research in the Nature Reserve of Aniene Valley and for their support in carrying out research within the park. We thank the reviewers for their constructive comments and suggestions, which substantially improved the manuscript.

**Conflicts of Interest:** The authors declare no conflict of interest.

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
