# Peer review of "Microplastics, Additives, and Plasticizers in Freshwater Bivalves: Preliminary Research of Biomonitoring"

_water, doi:10.3390/w15142647_

Round 1

Reviewer 1 Report

The article addresses a topic of great interest. The work is well structured and argued. Many new elements are introduced.

In my opinion the paper with title: “Microplastics, additives, and plasticizers in freshwater bivalves: preliminary research of biomonitoring could be published in the journal Water if the authors will make the corrections and specifications suggested:

1. Please enter the words: microplastics and Bivalves organisms under Key words

2. Please explain the meanings of the symbols used in the article. For example in figures 2 and 4, where the symbols of some polymers and plastic materials are shown.

3. Figures 2 and 4, in their current form, are not very suggestive, taking into account the colours used and the fact that they are not well distinguished. It is not clear what the identified polymers or materials are. The authors should present a clear form the figures.  

4.In chapter 2.5. please give details about the programs used for the statistical analyses. What relationships do these establish and between which parameters?

5. Please mention the significance of the parameters: χ2, df; etc.

Author Response

REV 1

The article addresses a topic of great interest. The work is well structured and argued. Many new elements are introduced.

In my opinion the paper with title: “Microplastics, additives, and plasticizers in freshwater bivalves: preliminary research of biomonitoring could be published in the journal Water if the authors will make the corrections and specifications suggested:

  1. Please enter the words: microplasticsand Bivalves organismsunder Key words

AUs: Thanks for the suggestion, we included these keywords.

  1. Please explain the meanings of the symbols used in the article. For example in figures 2 and 4, where the symbols of some polymers and plastic materials are shown.

AUs: We reported the information of the symbols presented in the main text as supplementary materials because the tables containing this information are lengthy and might make the text more difficult to follow. Specifically, Table S4 contains information on polymers, while Table S6 contains information on additives. We added in the caption of figures 2 and 4 the abbreviation of each polymers/additives, plasticizers, etc.

  1. Figures 2 and 4, in their current form, are not very suggestive, taking into account the colours used and the fact that they are not well distinguished. It is not clear what the identified polymers or materials are. The authors should present a clear form the figures.  

AUs: We agree with the Reviewer's feedback and have made revisions to the figures to enhance the clarity of the results. In relation to Figure 2, our objective was to highlight the polymer composition and the varying concentrations over time (ST and LT). Given this aim, we have opted to retain the same graph while incorporating distinct textures and acronyms to facilitate easy identification of the different polymers. In the case of Figure 4, since a clear quantitative trend over time was not apparent, we have created a new figure that illustrates the concentration of APFs both in percentage and total amount. We trust that we have addressed the Reviewer's concerns and enhanced the quality of the figures.

4.In chapter 2.5. please give details about the programs used for the statistical analyses. What relationships do these establish and between which parameters?

AUs: In lines 142-143 we specified the program used for the statistical analyses. We employed GraphPad Prism software (version 8.0.1). We clarified the relationships and parameters considered for the statistical analyses in lines 144-151.

  1. Please mention the significance of the parameters: χ2, df; etc.

AUs: We explained in detail all the symbol reported in the text in paragraph “2.5 Statistical analysis”.

Reviewer 2 Report

Dear Authors,

Thank you for your manuscript submission “Microplastics, additives, and plasticizers in freshwater bivalves: preliminary research of biomonitoring.” The manuscript studied the pollution of small plastics in three rivers in Italy using the Anodonta cygnea bivalve organisms as bioindicators. Here are my specific comments:

-          The innovation and significance of this study compared with previous studies should be mentioned in the introduction

-          Are there any assumptions for the experimental conditions and data analysis?

-          Line 198 (page 6), how did you get 1953 SMPs/g ww from figure 2?

-          Are there any limitations for the method in this study?

-          More details on future research should be indicated in the conclusion

Author Response

Thank you for your manuscript submission “Microplastics, additives, and plasticizers in freshwater bivalves: preliminary research of biomonitoring.” The manuscript studied the pollution of small plastics in three rivers in Italy using the Anodonta cygnea bivalve organisms as bioindicators. Here are my specific comments:

-          The innovation and significance of this study compared with previous studies should be mentioned in the introduction

AUs: Thanks for the observation, we better highlighted the innovation and significance provided by our research in lines 63-67. “Specifically, our study employs a novel method for investigating all present polymers without denaturation and analyses the gills and gastrointestinal tract separately, allowing for a more thorough evaluation of uptake and ingestion. These aspects represent important advancements in the analysis of plastic pollution and contribute to a deeper understanding of the MPs impacts on freshwater bivalves.”

-          Are there any assumptions for the experimental conditions and data analysis?

AUs: We clarified the assumptions for the experimental conditions and data analysis in paragraph 2.5. Our assumptions were that the concentration of SMPs and APFs would increase over time in the GIT and decrease in the gills. Additionally, we expected that in urban rivers, such as the Aniene, the concentration of SMPs and APFs would be higher compared to rivers with lower levels of urbanization. About data analysis, the Chi-square test (χ2) was used to compare the concentrations of SMPs and APFs in the same river at different times of exposure (ST vs LT) in both the gills and GITs. The Kruskal-Wallis test was then used to compare the differences in SMPs and APFs distributions among sites.

-          Line 198 (page 6), how did you get 1953 SMPs/g ww from figure 2?

AUs: In Figure 2A, we presented the values in thousands on the y-axis, with a value of 15 corresponding to 15,0000 SMPs/g. At T0, it can be observed that the value was between 19000 and 20000 SMPs/g in gills and about 2000 SMPs/g in GITs. Therefore, 1953 SMPs/g is the average value obtained from SMPs concentration in gills and GITs.

-          Are there any limitations for the method in this study?

AUs: The number of bivalves used in this study was limited due to the high costs of bivalve samples and time required for the exposure. Therefore, our research represents a preliminary study conducted to verify the feasibility of using this species for biomonitoring activities. We added this aspect in Conclusions paragraph. To address other critical aspects, we took all necessary precautions, including using a specific method to not denature polymers, selecting bivalves of similar size and age, and ensuring that physical-chemical parameters that could affect the filtration rate were similar between investigation sites.

-          More details on future research should be indicated in the conclusion

AUs: We added indication for future research and details about suggestion previously reported (see lines 346-347 and 359-362).

Reviewer 3 Report

Comments to the manuscript by Giulia Cesarini, Fabiana Corami, Beatrice Rosso and Massimiliano Scalici “Microplastics, additives, and plasticizers in freshwater bivalves: preliminary research of biomonitoring”

The paper is preliminary study, concerning accumulation of microplastics, plasticizers, and additives in the freshwater bivalves Anodonta cygnea through active biomonitoring.

I have the following comments to the paper.

1.      The number of studied organisms is too small. For example, dimensions of the sample Ctr3 smaller than those of Ctr1 and Ctr2, but wet gills weight of this sample more than twice larger than wet gills weight of two other samples. Dimensions of the samples in the group “Short term 1 month” vary within 10 percent but wet gills weight can be different up to 5 times (2.2 for Mar2 sample and 11.2 for Sac1 sample). These are data from the table S1. So, it is hard to decide what is typical.

2.      There are 18 different SMPs in the figure 2 legend, but in the histograms, it is difficult to distinguish the colors.

3.      There are more than 30 different APFs in the figure 4 legend, but it is hard to find that many colors at the histogram bars in the figure 4.

4.      The references [33,38] are not complete.

It seems important to clarify the comments to have the paper ready for publishing.

Author Response

Comments to the manuscript by Giulia Cesarini, Fabiana Corami, Beatrice Rosso and Massimiliano Scalici “Microplastics, additives, and plasticizers in freshwater bivalves: preliminary research of biomonitoring”

The paper is preliminary study, concerning accumulation of microplastics, plasticizers, and additives in the freshwater bivalves Anodonta cygnea through active biomonitoring.

I have the following comments to the paper.

  1. The number of studied organisms is too small. For example, dimensions of the sample Ctr3 smaller than those of Ctr1 and Ctr2, but wet gills weight of this sample more than twice larger than wet gills weight of two other samples. Dimensions of the samples in the group “Short term 1 month” vary within 10 percent but wet gills weight can be different up to 5 times (2.2 for Mar2 sample and 11.2 for Sac1 sample). These are data from the table S1. So, it is hard to decide what is typical.

AUs: We agree with the Auditor’s observation, the number of organisms was limited due to the high costs of bivalve samples and time required for the exposure. Therefore, our research represents a preliminary study conducted to verify the feasibility of using this species for biomonitoring activities. We added this aspect in Conclusions paragraph. To address other critical aspects, we took all necessary precautions, including selecting bivalves of similar size and age, and ensuring that physical-chemical parameters that could affect the filtration rate were similar between investigation sites.

  1. There are 18 different SMPs in the figure 2 legend, but in the histograms, it is difficult to distinguish the colors.
  2. There are more than 30 different APFs in the figure 4 legend, but it is hard to find that many colors at the histogram bars in the figure 4.

AUs: We agree with the Reviewer's feedback and have made revisions to the figures to enhance the clarity of the results. In relation to Figure 2, our objective was to highlight the polymer composition and the varying concentrations over time (ST and LT). Given this aim, we have opted to retain the same graph while incorporating distinct textures and acronyms to facilitate easy identification of the different polymers. In the case of Figure 4, since a clear quantitative trend over time was not apparent, we have created a new figure that illustrates the concentration of APFs both in percentage and total amount. We trust that we have addressed the Reviewer's concerns and enhanced the quality of the figures.

  1. The references [33,38] are not complete.

AUs: Thank you for reporting, we have added the correct references.

Round 2

Reviewer 1 Report

In my opinion the paper with title: “Microplastics, additives, and plasticizers in freshwater bivalves: preliminary research of biomonitoring should be published in the journal Water”. The authors made the corrections in accordance with the requirements.

Author Response

Thanks the Reviewer for providing positive feedback about our manuscript and for contributing to improve its quality. 

Reviewer 3 Report

I think the paper can be accepted in the present form

Author Response

Thanks to the reviewer for the positive feedback about our manuscript and for contributing to improve its quality. We included the acknowledgments to Reviewers in the manuscript.